# Machine learning electronic structure methods based on the one-electron reduced density matrix

Xuecheng Shao [1] ✉, Lukas Paetow[1], Mark E. Tuckerman [2,3,4,5] ✉ & Michele Pavanello [1,6] ✉

The theorems of density functional theory (DFT) establish bijective maps between the local external potential of a many-body system and its electron density, wavefunction and, therefore, one-particle reduced density matrix. Building on this foundation, we show that machine learning models based on the one-electron reduced density matrix can be used to generate surrogate electronic structure methods. We generate surrogates of local and hybrid DFT, Hartree-Fock and full configuration interaction theories for systems ranging from small molecules such as water to more complex compounds like benzene and propanol. The surrogate models use the one-electron reduced density matrix as the central quantity to be learned. From the predicted density matrices, we show that either standard quantum chemistry or a second machine-learning model can be used to compute molecular observables, energies, and atomic forces. The surrogate models can generate essentially anything that a standard electronic structure method can, ranging from band gaps and Kohn-Sham orbitals to energy-conserving ab-initio molecular dynamics simulations and infrared spectra, which account for anharmonicity and thermal effects, without the need to employ computationally expensive algorithms such as self-consistent field theory. The algorithms are packaged in an efficient and easy to use Python code, QMLearn, accessible on popular platforms.

Computational models are routinely employed to predict molecular and material properties in lieu of or prior to performing costly experiments. They are also used to explain the complex electron and nuclear dynamics that underlie experimental observations[1]. When these computational strategies require the evaluation of the electronic structure of a system, they often become the computational bottleneck, lengthening the time to solution. Consequently, an important and timely goal is the development of approaches capable of providing the electronic structure of complex systems at reduced computational cost[2–5] or even bypassing electronic structure calculations altogether. This article focuses on achieving the latter by leveraging the power of machine learning (ML).

The standard use of ML methods is to target single quantities of interest, which are learned in terms of a few descriptors. Examples of this are predictions of the electronic energy (including those that apply the concept of "delta" learning)[6–9], dipole moments, and polarizabilities[10,11], to name a few. Such a modus operandi is not ideal. Should the target be a quantity for which the model has not been

[1]Department of Chemistry, Rutgers University, Newark, NJ 07102, USA. [2]Department of Chemistry, New York University, New York, NY 10003, USA. [3]Courant Institute of Mathematical Science, New York University, New York, NY 10003, USA. [4]Simons Center for Computational Physical Chemistry, New York University, New York, NY 10003, USA. [5]NYU-ECNU Center for Computational Chemistry at NYU Shanghai, 200062 Shanghai, China. [6]Department of Physics, Rutgers University, Newark, NJ 07102, USA. ✉e-mail: xuecheng.shao@rutgers.edu; mark.tuckerman@nyu.edu; m.pavanello@rutgers.edu

trained, then a new model needs to be trained to predict this quantity. A relevant example is the computation of infrared (IR) spectra, where both the spectral line positions as well as the intensities are needed. A typical ML model that learns the potential energy surface of a molecular system or a material can only predict the spectral line positions through a molecular dynamics simulation followed by analysis of the velocity autocorrelation function[12]. However, in order to predict intensities, the autocorrelation function of the dipole moment is needed. The usual procedure to set up two ML models—one for the energy surface and one for the dipole moment—is time-consuming and ultimately avoidable. It is worth noting that there are various developing and complementary techniques that involve learning Hamiltonian matrices with respect to localized orbitals using symmetrized 2-[13] or N-center[14] representations, or deep neural network representation[15]. These methods share some of the underlying motivations of our work. For further reading on this and related topics, we recommend a recent comprehensive review[16].

An ideal ML method should learn from quantities that are dense in the amount of information they hold. The most general and ideal ML model would learn the many-body electronic wavefunction. From the wavefunction, one could predict the potential energy surface, the dipole moment and any other quantity of interest. ML models of many-body wavefunctions[17,18] are becoming competitive against other accurate wavefunction solvers, such as Quantum Monte Carlo. However, the complexity involved in both computations and training data sizes[19,20] hampers their broad applicability. Luckily, rigorous and bijective maps from DFT[21,22] and Reduced Density Matrix Functional Theory (RDMFT)[23–26] can be exploited to shift the focus from the many-body wavefunction to other, related quantities such as the electron density, $\rho(\mathbf{r})$, or the $N$-electron reduced density matrix ($N$-rdm), $\hat{\gamma}_N$. Caution must be used, as the bijective nature of the maps considered in the case of RDMFT can only be guaranteed for non-degenerate ground states. When spin systems are considered, additional constraints and features for the ML model for the 1-rdm would be required[27,28].

The electron density can be learned in terms of local atom-centered descriptors[11,29,30]. While energies obtained using these methods can be accurate to around $1\,\mathrm{kcal\cdot mol^{-1}}$, the model densities usually have deviations from the target on the order of a few percent. However, even with these small deviations, the model densities cannot be used directly to evaluate ionic forces in a DFT algorithm. This is because self-consistent electron densities must be converged to within much tighter thresholds for accurate force calculations. Exploiting the Hohenberg and Kohn theorems[21], the electron density can be used as the target quantity in ML models where external potentials serve as features[31,32]. This recovers model electron densities that, even if not completely accurate, can still be successfully used as a feature to accurately learn the energy and forces of methods such as DFT and coupled cluster[7,32,33].

The 1-rdm is emerging as a powerful feature for describing quantum systems, even when non-local correlations are important. For example, Schmidt et al. show that approximate 1-rdm energy functionals can be constructed (even with the aid of ML) for the description of correlated bosonic systems[34]. Focusing on the 1-rdm instead of the electron density has several advantages, e.g., the ability to deliver expectation values of any one-electron operator, including non-multiplicative operators such as the kinetic energy, the exchange energy, and the corresponding non-local (Hartree–Fock (HF)) potential. Additionally, much as can be done for the electron density, formal functionals of the 1-rdm can be learned[35], such as that of the electronic energy or corresponding atomic forces. Therefore, models for the 1-rdm should significantly extend the scope of ML models compared to those that learn the density alone. Wetherell et al.[35] considered the possibility of learning the functional $\hat{\gamma}[\rho]$ from exactly solvable real-space model systems achieving promising results. To date, however,

there have been no attempts to learn $\hat{\gamma}[\rho]$ or $\hat{\gamma}[v]$ (as we do here) for real systems, such as molecules.

In this work, we take on the challenge of learning 1-rdms to such an accuracy that the predicted 1-rdm are essentially indistinguishable from those delivered by standard electronic structure software. We achieve this aim by representing external potentials and target 1-rdms in terms of their matrix elements over Gaussian-type Orbitals (GTOs) and devising an efficient generator of training sets. This allows us to deliver "surrogate electronic structure methods" that predict 1-rdms to then deliver useful quantities (e.g., energy, forces, band gaps, orbitals) that are as accurate and useful as those computed by standard electronic structure software.

In the following sections, we present examples of calculations performed using various surrogate electronic structure methods, ranging from DFT to full configuration interaction, after introducing the main algorithms. Our proof of concept consists of seven molecules from small to medium-sized, rigid and floppy, which we have chosen to demonstrate the uniqueness and novelty of our method.

## Results
### Learning rigorous maps from DFT and RDMFT
We aim at learning rigorous maps from DFT[21] and RDMFT[23] linking the 1-rdm (the full matrix for RDMFT or just the diagonal elements for DFT) with virtually any ground state property given as the expectation value of any operator. Specifically, the following two maps are considered (dropping subscripts for notational ease),

$$\hat{v} \rightarrow \hat{\gamma}, \qquad\qquad\qquad (\text{map } 1)$$

$$\hat{\gamma} \rightarrow E, F, \langle \hat{O} \rangle, \qquad\qquad (\text{map } 2)$$

where $\hat{v}$ is the external potential, $E$ and $F$ are the electronic energy and the corresponding atomic forces, respectively, and $\langle \hat{O} \rangle$ is the expectation value of the operator $\hat{O}$. We call the ML procedure for map 1 $\gamma$-learning, and that for map 2 $\gamma + \delta$-learning. We note that when DFT methods are considered, the Kohn-Sham 1-rdm is targeted.

The utility of a ML model for map 1 is evident, as the computationally costly steps in electronic structure solvers (such as the self-consistent field (SCF) procedure or even more complex algorithms for post-Hartree−Fock (post-HF) methods) are replaced by the ML model. For similar reasons, the utility of learning map 2 is also evident. When considering mean field methods, map 2 can either be learned or directly computed from predicted 1-rdms. In the following sections, we will consider these options and show that they both lead to equally successful outcomes (vide infra). For post-HF wavefunction-based methods, map 2 must be learned, as there are no pure functionals of the 1-rdm that deliver energy and forces for these methods.

In this work, we represent 1-rdms and external potentials in terms of GTOs. GTOs are useful and convenient. For example, expectation values are simply computed, e.g., the non-interacting kinetic energy, $T_s[\hat{\gamma}] = -\frac{1}{2}\mathrm{Tr}[\nabla_{\mathbf{r}'}^2 \gamma(\mathbf{r}, \mathbf{r}')] = \mathrm{Tr}[\hat{\gamma}\hat{t}]$ (where $\nabla_{\mathbf{r}'}$ is the gradient with respect to the $\mathbf{r}'$ variable, $t_{\mu\nu} = -\frac{1}{2}\langle \mu|\nabla^2|\nu\rangle$, where $\mu$ and $\nu$ are GTO indices). Another important positive consequence of using GTOs is the possibility to work in an internal reference frame, providing a straightforward framework for dealing with the rotational and translational degrees of freedom. For ML models, this has been a significant challenge[11,36,37].

### Learning map 1: $\gamma$-learning
Inspired by Brockherde et al.[31], we learn map 1 by supervised ML exploiting a kernel ridge regression (KRR),

$$\hat{\gamma}[\hat{v}] = \sum_i^{N_{\text{sample}}} \hat{\beta}_i K(\hat{v}_i, \hat{v}). \qquad (1)$$

In the above equation $K(\hat{v}_i, \hat{v}_j) = \text{Tr}[\hat{v}_i \hat{v}_j]$, $\{\hat{v}_i\}$ is a training set of size $N_{\text{sample}}$ of external potentials and $\hat{\beta}_i$ are KRR coefficients which, in this case, are matrices with leading dimension of the number of atomic orbitals (AOs). The $\hat{\beta}_i$ coefficients are determined by KRR through the standard method of inversion of the regularized kernel matrix leading to (in matrix notation) $\hat{\beta}_i = \sum_j [\mathbf{K} + \lambda \mathbf{I}]_{ij}^{-1} \hat{\gamma}_j$, where $\mathbf{K}_{ij} = K(\hat{v}_i, \hat{v}_j)$, $\mathbf{I}$ is the identity matrix, $\lambda$ a regularization hyperparameter, and $\{\hat{\gamma}_j\}$ the target 1-rdms from the training set. The 1-rdms predicted by $\gamma$-learning are denoted by $\hat{\gamma}^p$.

### Learning map 2: $\gamma + \delta$-learning

While in $\gamma$-learning external potentials serve as features and the 1-rdms are the targets, in $\gamma + \delta$-learning, due to the shifted focus to learning functionals of the 1-rdm, the features are the matrix elements of the 1-rdm. Targets can be the 1-rdm itself (producing predicted 1-rdms of higher accuracy, hereafter denoted by $\hat{\gamma}^p + \Delta\hat{\gamma}^p$), the electronic energies and associated nuclear forces (hereafter denoted by $E_p$ and $F_p$), or, in principle, any other expectation value with respect to the ground state wavefunction. Another difference among the two learning steps are the types of regression used. As we have seen, KRR is employed for $\gamma$-learning. For $\gamma + \delta$-learning, after some testing, we chose to use a regularized linear regression. A summary of $\gamma$- and $\gamma + \delta$-learning and the adopted nomenclature for the associated predicted quantities is given in Table 1.

To summarize, we introduced three models: $\text{QM}^{\text{ML}}[\gamma^p]$, $\text{QM}^{\text{ML}}[\gamma^p + \Delta\gamma^p]$, and $\text{QM}^{\text{ML}}$, where QM can be any quantum mechanical method that yields the 1-rdm. $\text{QM}^{\text{ML}}[\gamma^p]$ and $\text{QM}^{\text{ML}}[\gamma^p + \Delta\gamma^p]$ denote methods where all properties are calculated directly from the 1-rdm from map 1 and map 2, respectively. $\text{QM}^{\text{ML}}$ is equivalent to $\text{QM}^{\text{ML}}[\gamma^p + \Delta\gamma^p]$, except energy and forces are predicted directly using map 2.

Once the ML models are formulated, the next step is to develop a strategy for training them, which is the subject of the next section.

### Training and benchmark tests

$\gamma-$ and $\gamma + \delta$-learning require carefully generated training sets. A set of molecular geometries that samples a desired configuration space needs to be formulated. In a second step, for $\gamma$-learning external potential/1-rdm pairs of matrices over a GTO basis for each geometry in the training set are needed. For $\gamma + \delta$-learning, the electronic energy, atomic forces, and any additional molecular property are needed.

Separate training sets must be generated for each molecule and electronic structure method chosen, including any method-specific settings such as basis sets, charge, and spin multiplicity. This aspect of our method can be seen as both an advantage and a limitation. On one hand, it allows for flexibility in choosing the electronic structure method best suited for a particular application. On the other hand, it means that in principle, a new ML model should be trained for each combination of settings. However, our results in the next section show that the limitations of the method are mitigated by its potentially good transferability.

Key to the success of this effort is to limit the training set to the smallest number of elements possible. Our guiding principle is to train the ML models, carrying out only a fraction of the simulations needed

to run a standard ab initio molecular dynamics (AIMD) simulation with the target electronic structure method.

Such a tall order can be achieved using a sampling method based on vibrational normal mode analysis. For each molecule, we (1) carry out a normal mode analysis at the equilibrium geometry, and (2) generate $N_{\text{sample}}$ geometries randomly displaced from the equilibrium geometry along each normal mode according to a normal (Gaussian) distribution of variance $\sigma_i^2 = 2k_B T N_{\text{atoms}} / [\Omega_i^2 N_{\text{vib}} (1 - 2/9N_{\text{vib}})^3]$, where $T$ is a target temperature, $N_{\text{vib}}$ is the number of vibrational degrees of freedom, and $N_{\text{atoms}}$ is the number of atoms, and $\Omega_i$ is the vibrational frequency of mode $i$. In principle, sampling $N_{\text{vib}}$ modes with $N_s$ points for each mode would lead to $N_{\text{sample}} = (N_s)^{N_{\text{vib}}}$ total sampled points, which would be unattainable. However, after some testing, we noticed that $N_{\text{sample}} \propto N_{\text{vib}}^3$ is more than sufficient to achieve a deviation of about $0.1\,\text{kcal}\cdot\text{mol}^{-1}$ for the predicted energy. For floppy molecules we carry out samplings from several initial geometries that reflect the number of stable conformers.

As shown in Table 2, the training set size, $N_{\text{sample}}$, is 27 (or $N_{\text{sample}} = N_{\text{vib}}^3$) for molecules with $N_{\text{vib}} = 3$, and it grows to 13,824 for benzene which has $N_{\text{vib}} = 30$ (i.e., $N_{\text{sample}} = (0.8 \cdot N_{\text{vib}})^3$). $N_{\text{sample}}$ then triples for methanol, 1- and 2-propanol due to the need to generate samples for the three most stable conformers (found by rotating the H atom of the OH group around the axis given by the C–O bond).

Supplementary Fig. S1 shows histograms of the electronic energies corresponding to the sampled geometries for benzene in comparison to an AIMD simulation. The sampling method captures, within a margin of error, the electronic energy distribution with a much-reduced number of energy evaluations compared to the AIMD simulation. In Supplementary Fig. S2, we show that the distribution of geometries in each normal mode is also correctly recovered. The distributions are given in the energy-scaled displacement coordinates (i.e., it is expected that all distributions share the same energy-scaled variance, $\sigma^2 = \sigma_i^2 \Omega_i^2$).

The goal of our ML model is to achieve accuracies beyond chemical accuracy, as our model aims to predict the 1-rdm with an accuracy that matches the SCF densities produced by conventional quantum chemistry codes. SCF densities are typically converged to energy thresholds of $10^{-5}$ or $10^{-6}$ Hartree, which is two to three orders of magnitude more accurate than chemical accuracy. In Fig. 1, we show the energy accuracy achieved for the total energy of benzene using training sets with varying numbers of structures. Chemical accuracy is already achieved with a training set size of 30 structures, and a one-order-of-magnitude improvement can be achieved by increasing the training set size to 100. However, to achieve accuracies three orders of magnitude below chemical accuracy and match the accuracy of conventional SCF-derived densities, larger training sets are required, as reported above.

Table 2 reports results for a surrogate of DFT within the local-density approximation (LDA). The conventional method is indicated by LDA and the surrogate model by $\text{LDA}^{\text{ML}}$. We compute and predict quantities along a 10 ps AIMD for several molecular systems, from small and rigid (such as water) to medium size and floppy (such as 1- and 2-propanol).

Energies, forces, dipoles, and non-interacting kinetic energies can either be predicted through separate regressions using $\gamma + \delta$-learning, or calculated from the predicted 1-rdm obtained from $\gamma$-learning (i.e., with $\hat{\gamma}^p$) or $\gamma + \delta$-learning (i.e., with $\hat{\gamma}^p + \Delta\hat{\gamma}^p$). In Table 2, dipole moments and non-interacting kinetic energies are only computed from the predicted 1-rdms. When employing the most accurate model, dipole deviations are at most $10^{-4}$ Debye (D) per vibrational degree of freedom, and non-interacting kinetic energies deviate by $1\,\text{kcal}\cdot\text{mol}^{-1}$.

Table 2 also shows that energies and forces computed by the 1-rdm from $\gamma + \delta$-learning (indicated by the predicted 1-rdm, $\hat{\gamma}^p + \Delta\hat{\gamma}^p$), substantially improve the result from $\gamma$-learning. While the electronic

**Table 1 | Brief summary of the machine-learning (ML) models employed in this work: $\gamma$-learning and $\gamma + \delta$-learning**

|  | Model | Features | Targets | Prediction |
|---|---|---|---|---|
| $\gamma$-learning | Eq. (1) | $\hat{v}$ | $\hat{\gamma}$ | $\hat{\gamma}^p$ |
| $\gamma + \delta$-learning | rLR | $\hat{\gamma}^p$ | $\hat{\gamma}, E, F$ | $\hat{\gamma}^p + \Delta\hat{\gamma}^p, E_p, F_p$ |

$\hat{v}$ and $\hat{\gamma}$ stands for external potential and one-electron reduced density matrix, respectively. $E$ and $F$ stand for electronic energy and atomic forces, respectively. The subscripts and superscripts $p$ indicate the predicted quantities. The row "Predicted" introduces the nomenclature of the quantities predicted by the ML methods. rLR stands for regularized linear regression.

**Table 2 | Benchmark study for the surrogate electronic structure method for the local-density approximation (LDA), LDA$^{ML}$**

| System | | H$_2$O | CO$_2$ | NH$_3$ | CH$_3$OH* | C$_6$H$_6$ | 1-propanol* | 2-propanol* |
|---|---|---|---|---|---|---|---|---|
| $N_{vib}$ | | 3 | 3 | 6 | 12 | 30 | 30 | 30 |
| Training set size ($N_{sample}$) | | 27 | 27 | 216 | 5184 | 13,824 | 41,472 | 41,472 |
| Energy (kcal · mol$^{-1}$) | LDA$^{ML}[\gamma^p]$ | 0.0004 | 0.0020 | 0.0008 | 0.0050 | 0.0145 | 0.1640 | 0.0636 |
| | LDA$^{ML}[\gamma^p + \Delta\gamma^p]$ | 0.0003 | 0.0020 | 0.0008 | 0.0014 | 0.0055 | 0.0270 | 0.0076 |
| | LDA$^{ML}$ | 0.0233 | 0.0281 | 0.0037 | 0.0117 | 0.0143 | 0.1580 | 0.1289 |
| Force (kcal · mol$^{-1}$ · Å$^{-1}$) | LDA$^{ML}[\gamma^p]$ | 1.53 | 2.56 | 1.74 | 2.79 | 3.78 | 5.40 | 3.67 |
| | LDA$^{ML}[\gamma^p + \Delta\gamma^p]$ | 0.53 | 0.24 | 0.15 | 1.05 | 1.05 | 2.35 | 1.03 |
| | LDA$^{ML}$ | 0.09 | 0.08 | 0.01 | 0.07 | 0.05 | 1.16 | 0.41 |
| Dipole (10$^{-3}$ D · $N_{vib}^{-1}$) | LDA$^{ML}[\gamma^p]$ | 0.57 | 0.25 | 0.19 | 0.13 | 0.13 | 0.35 | 0.35 |
| | LDA$^{ML}[\gamma^p + \Delta\gamma^p]$ | 0.05 | 0.06 | 0.02 | 0.05 | 0.02 | 0.16 | 0.08 |
| Kinetic Energy (kcal · mol$^{-1}$) | LDA$^{ML}[\gamma^p]$ | 0.28 | 0.61 | 0.56 | 1.46 | 1.12 | 5.42 | 4.14 |
| | LDA$^{ML}[\gamma^p + \Delta\gamma^p]$ | 0.11 | 0.05 | 0.03 | 0.19 | 0.35 | 1.09 | 1.00 |

Root-mean-square deviations (RMSDs) of predicted energy, magnitude of the atomic forces, magnitude of the dipole moment vector, and the non-interacting kinetic energy, along a 10 picosecond (ps) ab initio molecular dynamics trajectory sampled every 100 femtoseconds (100 test structures in total) at 300 K. LDA$^{ML}[\gamma^p]$ and LDA$^{ML}[\gamma^p + \Delta\gamma^p]$ compute all quantities from the predicted one-electron reduced density matrices $\hat{\gamma}^p$ and $\hat{\gamma}^p + \Delta\hat{\gamma}^p$. $N_{vib}$ is the number of vibrational degrees of freedom in the molecule. The superscript * indicates that samples are generated from three stable conformers.

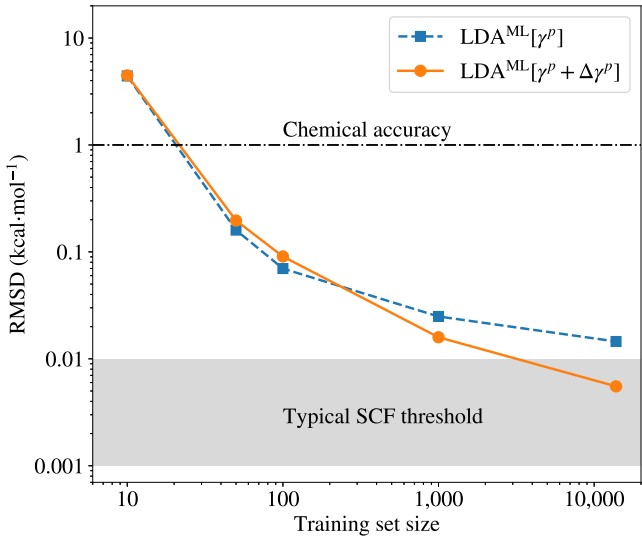

**Fig. 1 | Accuracy of the surrogate electronic structure methods as a function of training set size for the benzene molecule.** The root-mean-square deviation (RMSD) of calculated energies from LDA$^{ML}[\gamma^p]$ and LDA$^{ML}[\gamma^p + \Delta\gamma^p]$ surrogate models of the local-density approximation (LDA) from predicted one-electron reduced density matrices $\hat{\gamma}^p$ and $\hat{\gamma}^p + \Delta\hat{\gamma}^p$ are presented. The typical energy convergence thresholds of self-consistent field (SCF) methods are indicated by the shadowed region.

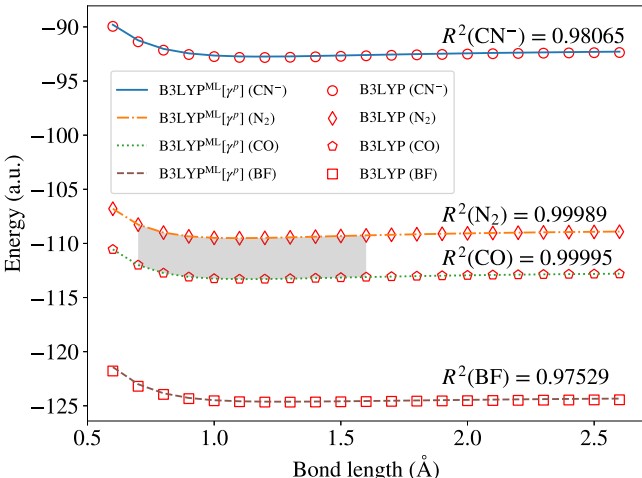

**Fig. 2 | Total energy as a function of bond length for the isoelectronic series N$_2$, CO, CN$^-$, and BF.** The training set is composed only of N$_2$ and CO geometries with bond lengths indicated by the shadowed region. Hybrid functional B3LYP benchmark results are shown by scatter points, and our surrogate model results are represented by lines. The coefficient of determinations ($R^2$) values for different molecules are also represented.

energies do not improve much, the computed atomic forces improve more, deviating by not bigger than 3 kcal · mol$^{-1}$ · Å$^{-1}$. Such a deviation is acceptable for AIMD and geometry optimization. As expected, computed expectation values, i.e., dipoles and non-interacting kinetic energy are also improved when they are evaluated with $\hat{\gamma}^p + \Delta\hat{\gamma}^p$ rather than just with $\hat{\gamma}^p$.

In Supplementary Table S1, we present the RMSDs for individual terms in the energy functional. It is observed that the RMSDs of these separate terms are larger compared to the RMSD for the total energy. This suggests that the model is benefiting from some degree of error cancellation. Notably, however, our model can accurately predict HF exchange energies using the predicted 1-rdms (with an RMSD of less than 0.2 kcal · mol$^{-1}$).

After testing (see Supplementary Table S2 for RMSDs of the 1-rdm's occupation numbers), we observed that strict non-interacting N-

representability of the 1-rdms is crucial for accessing stable, long AIMD trajectories for molecules that are larger than triatomics. To achieve this, we compute the occupation numbers of the predicted 1-rdms and enforce the Aufbau occupation. However, we omit this purification step when learning full configuration interaction (Full-CI) 1-rdms. In section III, we will also test the ability of the model 1-rdms to predict fractional occupations upon bond breaking.

While a thorough analysis of the transferability properties of our ML models is beyond the scope of this work, we tested the ability of the model trained on two molecular species, CO and N$_2$, together to predict two systems not included in the training set, the CN$^-$ anion and BF. Training and test molecules are isoelectronic and share the same number of AOs within the cc-pVTZ basis set which was chosen for this test. The results are displayed in Fig. 2 and show that the ML model can predict the energy of CN$^-$ and BF vs bond length quite well with a coefficient of determination ($R^2$) value of 0.9807 and 0.9753, respectively. For CO and N$_2$ (which are part of the training set) the $R^2$ score is 0.9999.

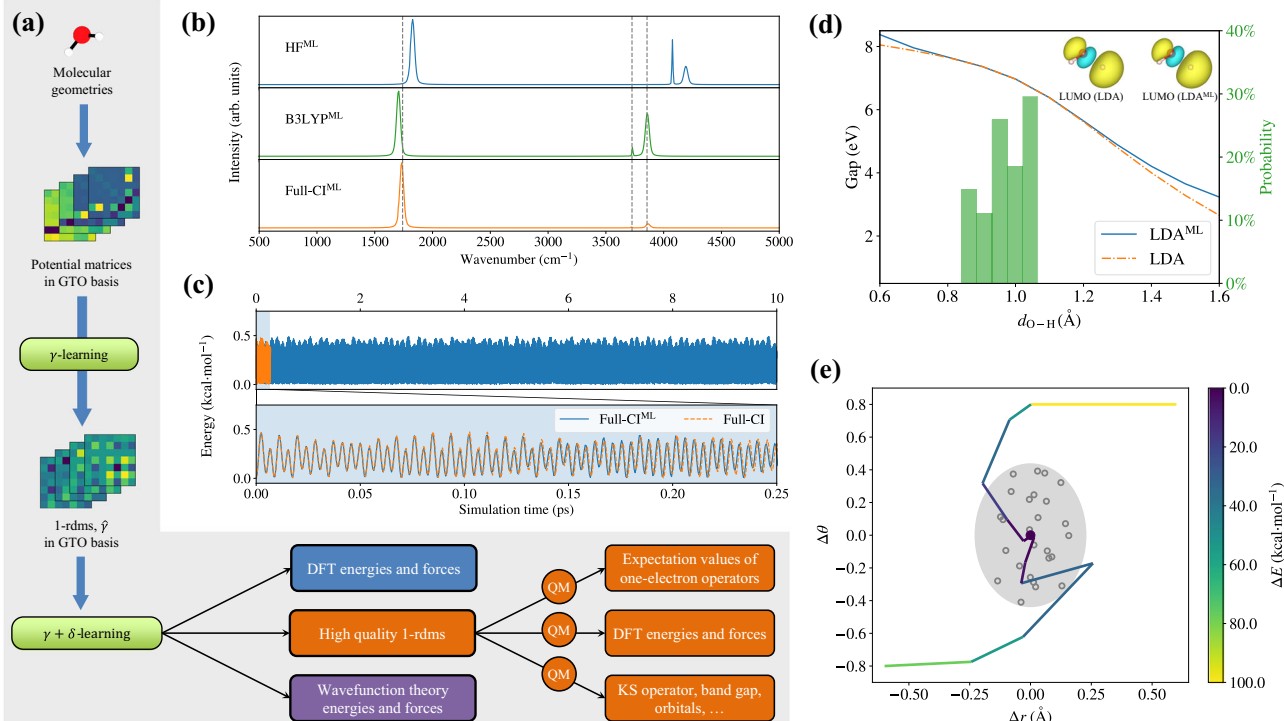

**Fig. 3 | Performance of the surrogate electronic structure methods (superscript ML) for water. a** The machine-learning methods used to make predictions are described. For each molecular geometry, the matrix elements of the external potential (e.g., electron-nuclear attraction potential) with respect to a Gaussian-type Orbital (GTO) basis set are computed. $\gamma$-learning uses one-electron reduced density matrices (1-rdms) as targets in the GTO basis via Eq. (1), exploiting map 1. $\gamma + \delta$-learning targets electronic energies and forces from any desired electronic structure method, and further refines the 1-rdm itself, exploiting map 2. **b** The predicted infrared spectra of water are compared to coupled-cluster singles and doubles with perturbative triples (CCSD(T)) Car-Parrinello molecular dynamics vibrational frequencies which are taken to be our benchmark, shown with dashed lines in the figure. **c** The electronic energy along one ab initio molecular dynamics trajectory with initial random velocities consistent with a temperature of 300 K in

the microcanonical ensemble is shown. **d** The highest occupied molecular orbital (HOMO)-lowest unoccupied molecular orbital (LUMO) gap as a function of the O–H distance ($d_{O-H}$) is plotted, along with the associated LUMO orbital for $d_{O-H} = 1.6$ Å, compared to the conventional local-density approximation (LDA) results. The training set structures are indicated by a green histogram. For LDA$^{ML}$ surrogate model, the HOMO-LUMO gaps and orbitals are computed with predicted 1-rdms from map 2. **e** The geometry relaxations of $H_2O$ molecules are carried out with the surrogate model of full configuration interaction (Full-CI$^{ML}$). The O–H distance ($\Delta r$ in Å) and H–O–H angle ($\Delta \theta$ in radians) referenced to the equilibrium values are shown. The training set structures are indicated by empty gray circles and the equilibrium geometry is represented by a dark red dot. The light gray area indicates the configuration space covered by the training set.

## The QMLearn software and workflow

The methods developed in this work are collected in the all-Python QMLearn software which is freely available from GitLab[38] and easily installed through `pip install qmlearn`. QMLearn is composed of the following classes: (1) a database collecting the training sets; (2) QM engines (we use PySCF, although other engines are also supported) capable of generating the training sets and matrix elements over the GTOs as well as the needed infrastructure to compute energies and atomic forces; (3) a structure handler (we use Atomistic Simulation Environment (ASE)[39]). ASE is used to handle molecular geometry, including driving molecular dynamics simulations; and (4) ML modules such as scikit-learn[40] or Tensorflow[41] (the current version of QMLearn supports only scikit-learn).

## Showcasing several surrogate electronic structure methods for water

Figure 3 showcases the performance of several surrogate electronic structure methods for the water molecule. Water is a small molecule, with only three normal modes of vibration, it is feasible to run Full-CI simulations with the 6-31G* basis set and train the surrogate Full-CI$^{ML}$ over a training set of only 27 structures. In addition to Full-CI, we also developed surrogates for Hartree−Fock, DFT within the LDA, and DFT using the hybrid B3LYP exchange-correlation (xc) functional. We tested the surrogate models in such common tasks as the computation of band gaps as a function of varying molecular geometry, the shape of

LUMO orbital, AIMD simulations in the microcanonical ensemble (constant number of particles, $N$, volume, $V$, and total energy, $E$, denoted hereafter as NVE), the prediction of IR spectra, and geometry relaxation.

For an AIMD simulation, Full-CI$^{ML}$ is stable for tens of picoseconds and produces energies that are very close to those of a conventional Full-CI simulation (which could be run for a shorter simulation time and is found to be four orders of magnitude slower than Full-CI$^{ML}$). When the IR spectrum is calculated using DFT surrogates, it deviates somewhat from the coupled-cluster singles and doubles with perturbative triples (CCSD(T)) benchmark of Ref. 42. As expected, when the IR spectrum is calculated using Full-CI$^{ML}$, it matches the benchmark's vibrational frequencies.

To evaluate the performance of our model, Full-CI$^{ML}$ 1-rdms in predicting fractional occupations upon bond breaking, we extended the water molecule training set with two additional dissociation geometries, $H_2O \rightarrow OH + H$ and $H_2O \rightarrow O + H_2$. As shown in Supplementary Figs. S3 and S4, the model 1-rdms are able to smoothly interpolate between Aufbau occupation states at the equilibrium geometry and fractional occupations at the dissociation limits. However, for the Full-CI occupations of $O + H_2$, a sharp transition is observed associated with the breaking of the O−H bonds and formation of the H−H bond, which is only smoothly interpolated by the model 1-rdms. Despite this limitation, the dipole moment vectors in the two dissociation limits are qualitatively reproduced by the model. Overall, these results suggest

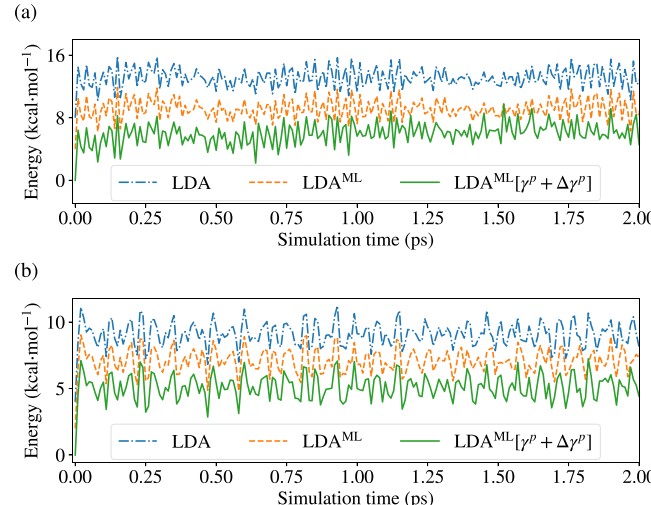

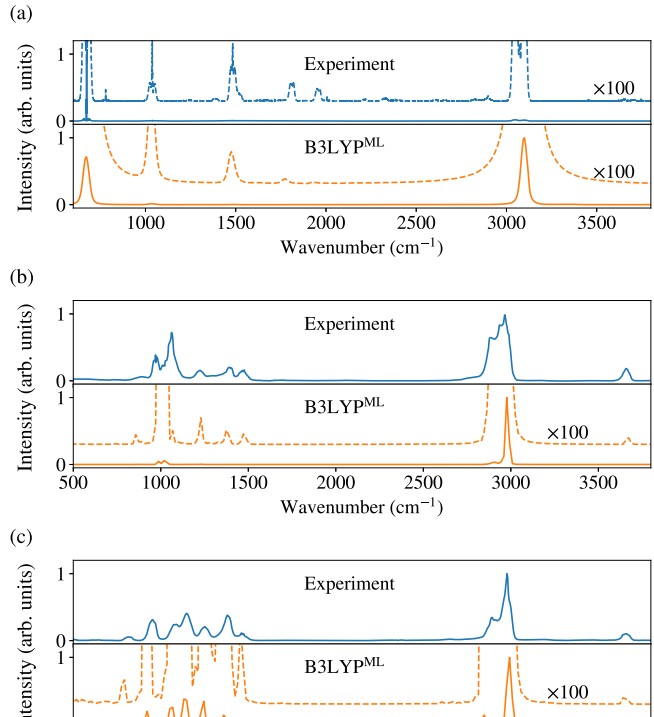

**Fig. 4 | Potential energies from conventional and surrogate electronic structure methods.** Electronic energy values along three independent ab initio molecular dynamics trajectories for **a** benzene and **b** 1-propanol carried out in the microcanonical (NVE) ensemble with initial velocities corresponding to an instantaneous temperature of 300 K. The three trajectories share the same initial conditions but are run with three distinct methods: local-density approximation (LDA) benchmark, machine-learning (ML) surrogate models, $LDA^{ML}$ and $LDA^{ML}[\gamma^p + \Delta\gamma^p]$ (which uses forces and energies computed from the predicted one-electron reduced density matrix $\hat{\gamma}^p + \Delta\hat{\gamma}^p$). We only report energies from snapshots taken every 10 femtoseconds of simulation time. To aid visualization, $LDA^{ML}$ and $LDA^{ML}[\gamma^p + \Delta\gamma^p]$ energies are shifted by 4 and 8 kcal · mol⁻¹, respectively, for (**a**) and by 2 and 4 kcal · mol⁻¹, respectively, for (**b**).

**Fig. 5 | Predicted gas-phase infrared spectra with surrogate electronic structure methods.** The $B3LYP^{ML}$ surrogate model is used for **a** benzene, **b** 1-propanol and **c** 2-propanol. Gas-phase experimental spectra from the National Institute of Standards and Technology Standard Reference Database[50] are reproduced for comparison. A vibrational scaling factor of 0.97 was applied to the computed spectra. For ease of visualization, we also feature magnified intensities by a factor of 100. Here we use 6-311G* basis set for 1- and 2-propanol.

that bond breaking can be qualitatively described by providing the ML model with critical information about the dissociation limits. To achieve a more accurate description of bond breaking, we anticipate that increasing the size of the training set will lead to progressively more accurate model 1-rdms.

The bandgap from the surrogate LDA DFT method, $LDA^{ML}$, follows closely the conventional LDA result; the orbitals do as well and are very similar to LDA. A similarly accurate result is found for geometry relaxations, where the equilibrium geometry is found starting from structures that are far away from the training set.

We therefore conclude that the surrogate models considered for the water molecule deliver a predicted electronic structure that is extremely close to the target electronic structure method, even for molecular geometries that are well outside the configuration space spanned by the training set.

## Ab initio dynamics and IR spectra

A more stringent test of the robustness of the surrogate electronic structure methods is the generation of fully deterministic energy-conserving dynamics. Figure 4 reports the electronic energies along NVE trajectories (the equilibrium geometry was used as initial geometry and the same random initial velocities were used for the methods reported) at an instantaneous temperature of 300 K for benzene and 1-propanol. Two flavors of the $LDA^{ML}$ surrogate model are considered: $LDA^{ML}$ and $LDA^{ML}[\gamma^p + \Delta\gamma^p]$. The latter only predicts 1-rdms (the $\hat{\gamma}^p + \Delta\hat{\gamma}^p$) and computes energies and forces from them.

Using $LDA^{ML}[\gamma^p + \Delta\gamma^p]$ and associated forces or the predicted energies and forces for $LDA^{ML}$ leads to essentially equivalent results. Specifically, we see that trajectories from the LDA surrogates follow closely the conventional LDA result up to a simulation time of 1 ps, after which they begin to slightly deviate. This behavior is in line with the RMSDs presented in Table 2.

The surrogate electronic structure methods are able to predict, for example, molecular dipole moments along a AIMD trajectory

directly from the predicted 1-rdms (i.e., the $\hat{\gamma}^p + \Delta\hat{\gamma}^p$). This can be exploited to compute anharmonic and temperature-dependent spectra of molecules[12].

Figure 3a already reports the IR spectrum of a water molecule. Moving on to larger and more floppy molecular systems, in Fig. 5 we report the gas-phase IR spectra of benzene, 1- and 2-propanol computed by the surrogate $B3LYP^{ML}$ method. Given the large number of vibrational normal modes, for each of the three molecules, we first run a 10 ps NVT dynamics at 300 K with $B3LYP^{ML}$. We then selected five snapshots for benzene and twelve snapshots for 1- and 2-propanol (sampled between 5 and 10 ps of the trajectory) and run additional 2 ps NVE trajectories computing the dipole moment at each step. IR spectra are computed by Fourier transformation of the dipole autocorrelation function. The model successfully reproduces the peak positions of the experimental spectra and follows the correct trend for peak intensities. However, due to the flexible nature of the molecules, particularly 1-propanol, a greater number of snapshots and longer trajectories may be necessary to obtain a more accurate comparison to the experiment. More importantly, the use of the B3LYP xc functional, which is an approximation, introduces errors in the predicted IR spectra, affecting both the frequency positions and the oscillator strengths[43,44]. Our $B3LYP^{ML}$ model excellently reproduces the B3LYP dipoles along one of the AIMD trajectories used for the computation of the IR spectra (see, e.g., Supplementary Fig. S5), showing that the deviations from the experiment are likely due to the approximations involved in the B3LYP xc functional.

## Predicted HOMO-LUMO gap and orbitals

DFT methods provide the ability to calculate band gaps and generate HOMO and LUMO orbitals. This is useful for a variety of purposes,

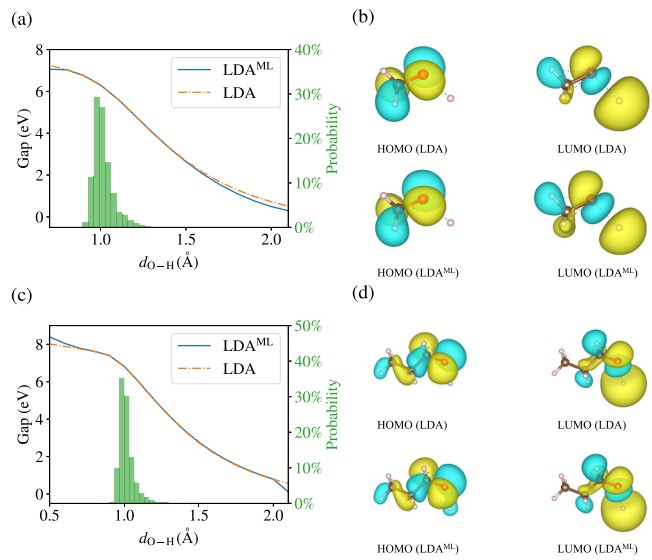

**Fig. 6 | Predicted band gaps and orbitals. a** The highest occupied molecular orbital (HOMO)–lowest unoccupied molecular orbital (LUMO) gap as a function of the O–H distance ($d_{O–H}$) for methanol. The green histogram is the probability distribution of the O–H distance for the training set. **b** HOMO and LUMO orbital isosurfaces from local-density approximation (LDA) and surrogate model LDA$^{ML}$ at the O–H distance of $d_{O·H}$ = 2.0 Å for methanol. **c, d** are same as (**a, b**) but for 1-propanol.

including the interpretation of photophysics and reactivity[45]. We therefore found it important to showcase the ability of the surrogate models to predict band gaps and orbitals for molecules other than water and for geometries near and far from the configuration space spanned by the training set. In addition to the already discussed Fig. 3d, Fig. 6 considers methanol and 1-propanol where the O–H bond is artificially shortened and stretched in the range 0.7–2.1 Å and 0.5–2.1 Å, respectively. The predicted LDA$^{ML}$ HOMO-LUMO gaps are overall in excellent agreement with the LDA results for both molecules. Similar to what we witnessed for water, LDA vs LDA$^{ML}$ gaps start to deviate only for geometries far away from the training set, see green histogram of O–H bond distances from the training set.

To appreciate the power of these surrogate models, we also reproduced with isosurface plots the predicted HOMO and LUMO orbitals for the most stretched configuration (O–H bond length of 2 Å). The results are similar to what we presented for water, i.e., they show that the orbitals from surrogate models are correct even for geometries very far away from the configuration space spanned by the training set.

## Discussion
We developed surrogate electronic structure methods based on machine learning of the one-electron reduced density matrix, or 1-rdm. When given a target molecule, the surrogate methods provide the same information and predictions as traditional electronic structure software.

We showcased DFT, HF, and post-HF surrogates for rigid molecules such as water and benzene and for flexible molecules such as 1- and 2-propanol. First, our models learned the 1-rdm of these systems as a function of the external potential. Then, rigorous maps were machine-learned from DFT and RDMFT to predict the 1-rdm to energy and 1-rdm to atomic forces maps.

Our surrogate methods robustly predict geometry optimizations, ab initio dynamics, and IR spectra from the molecular dipole moment. Because of their versatility, the surrogate methods predict not only structure and dynamics but also expectation values of one-electron operators and Kohn-Sham orbitals. We predicted HOMO and LUMO

orbitals and energy gaps for several molecules, with results very close to traditional methods even for geometries far away from the configuration space sampled by the training sets.

This work is a proof of concept showing that surrogate electronic structure methods can replace conventional electronic structure methods for most computational chemistry tasks. However, the extending the algorithm to higher-order rdms would enable the computation of expectation values of two-electron operators, energies and forces and will be considered in the future. Condensed phase systems and larger molecules are also targeted for future development. To approach large molecules, initial guesses of the 1-rdms can be constructed combining the 1-rdms of separate molecular fragments learned from smaller molecules. Their coupling can then be learned separately.

## Methods
### Electronic structure calculations
We generate training sets and to compute all the needed matrix elements over the AOs with PySCF[46]. DFT calculations carried out with the LDA xc functional use the parametrization from Perdew and Zunger[47]. Unless otherwise stated, we use the cc-pVTZ basis set for all systems except benzene, 1– and 2-propanol for which we use the 6-31G* basis set.

ASE[39] is employed to drive all AIMD simulations and geometry optimizations. Regressions are carried out with scikit-learn[40].

### Computation of IR spectra
IR spectra of the gas-phase water molecule were computed by Fourier transform of the molecular dipole moment along AIMD trajectories carried out in the NVE ensemble. Three trajectories were considered, one for each of the molecular vibrational normal modes. The initial geometry was taken to be the equilibrium geometry, and the initial velocities for each trajectory were proportional to the normal mode vector with a proportionality constant, "kick strength", $\varepsilon_i$, chosen to be compatible with an initial instantaneous temperature of 50 K for each mode. For the other molecules, given a large number of normal modes, we opted for a standard sampling method described in the main text.

In all cases, for each step in the trajectory, we represented the molecular coordinates in the so-called Eckart frame. The dipole moment in this frame, was then processed. First, the derivative of the dipole moment is computed, $\dot{\mu}(t)$. Then, the autocorrelation function of the dipole derivative was computed, and Fourier transformed to give the IR spectrum.

## Data availability
Training and test sets, as well as notebooks that fully reproduce all figures and all tables in this study, have been deposited on Zenodo[48].

## Code availability
The QMLearn software used for this study is available on GitLab at https://gitlab.com/pavanello-research-group/qmlearn. A snapshot of the code is also available on Zenodo[49].

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

## Acknowledgements

This material is based upon work supported by the National Science Foundation under Grants No. CHE-1553993, CHE-2154760, OAC-1931473, and Petroleum Research Fund grant No. 62555-ND6. M.E.T. acknowledges support from the Camille and Henry Dreyfus Foundation grant no. ML-22-146. We thank the Office of Advanced Research Computing at Rutgers for providing access to the Amarel cluster.

## Author contributions

X.S. wrote the code, developed the sampling method and analyzed the data. X.S. and L.P. performed simulations. M.P. and M.E.T. co-supervised the project. M.P. proposed the initial idea. All authors co-wrote the manuscript.

## Competing interests

The authors declare no competing interests.
