## [Peer Review File · Nature Communications]

REVIEWER COMMENTS

Reviewer #1 (Remarks to the Author):

Reviewer comments on "Machine Learning Electronic Structure Methods Based On The One-Electron Reduced Density Matrix."

In this work, the authors employ a simple but effective machine learning (ML) procedure to reproduce/predict the 1-rdm avoiding the need for its computation from first-principles calculations. To that end, they represent the 1-rdm and the external potential in the Gaussian Type Orbital[GTO] basis set using the latter to define the features of the ML procedure. Then, they use/propose a second ML procedure to either improve the 1-rdm, or compute quantities of interest from it (e.g. Forces, Dipoles, HOMO-LUMO gaps, etc).

The paper is well-written and very easy to follow. But, there are some issues that I would like to highlight.

1) In the abstract, the authors mention that the theorems of DFT and RDMFT establish a bijective map between the external potential of many-body systems and its electron density of 1-rdm. This statement is incorrect because in his paper Gilbert explicitly states that the one-to-one correspondence between the 1-rdm and a general non-local external potential is not possible. Thus, the authors should clearly mention that the bijective relation holds only for local potentials (same in the first paragraph in Pag. 2).

2) The introduction section misses a proper description of the state-of-the-art of ML procedures in the DFT and RMFT context. For example, citations to previous works are missing. The authors should cite the previous works done by M. Rupp, A. Tkatchenko, O.A. Von Lilienfeld, K. Burke, S. Luber, M.A.L. Marques, etc. Also, the previous works using ML in the RDMFT context should be introduced and cited, in particular,

Schmidt, Jonathan, Matteo Fadel, and Carlos L. Benavides-Riveros. "Machine learning universal bosonic functionals." *Physical Review Research* 3.3 (2021): L032063.

and

Wetherell, Jack, et al. "Insights into one-body density matrices using deep learning." *Faraday Discussions* 224 (2020): 265-291.

3) The AIMD acronym is never defined.

4) The first ML procedure (labeled as map 1) aims to avoid the computation of the 1-rdm from an expensive electronic structure calculation. The transferability of this procedure raises my major concern related to this work. I understand that for each system in table II the authors performed a KRR using some samples (N_{samples}). This procedure will definitely lower the computational cost during a MD simulation. But, if one uses the cc-pVTZ basis set, for example, and performs the KRR regression for N_2 with LDA (as the authors did for H_2O , CO_2 , etc.). Would the authors be able to predict the 1-rdm for a given geometry of CO by building the corresponding potential in the GTO basis and evaluating it in the KRR obtained for N_2 ? (Note: N_2 and CO in the cc-pVTZ basis set produce the same number of GTOs; also, both systems contain 14 electrons). The transferability test of the map 1 ML procedure should be addressed and discussed (also including other examples).

5) My second concern is related to spin-states. For different spin states the external potential represented in the GTO basis set is the same, but the 1-rdm are (in general) very different. This limitation of their ML procedure should be clearly mentioned in the manuscript.

6) The authors reported in Table II the RMSDs for some quantities. The larger errors in the Kinetic energy w.r.t. total Energies make me consider that the non-locality of the 1-rdm might be hard to reproduce by their ML procedure. For DFT, the authors could include, at least in the supporting information, all the energy components (i.e. the Kinetic energy, the interaction with the external potential, V_{Hartree} , V_{xc} , etc.). Actually, for the Hartree-Fock tests, the exchange energy could also serve as an excellent tool to evaluate the quality of the non-local character of the predicted 1-rdms.

7) In line with the previous point, it is important to test the N-representability of reduced density matrices. In the KS-DFT/HF cases, the 1-rdm must correspond to that of a single-determinant wave function. Thus, upon diagonalization of the 1-rdm (probably diagonalizing $S^{1/2} P S^{1/2}$) the authors could produce the occupation numbers and discuss whether or not their ML procedure is able to produce predicted 1-rdms that correspond to a single-determinant wave function. At the FCI level, similar tests should be performed considering that occupation numbers should yield between 0 and 1 (see C. Garrod and J. K. Percus *J. Math. Phys.* 5, 1756 (1964) <https://aip.scitation.org/doi/pdf/10.1063/1.1704098>). This test could reveal if the ML procedure proposed by the authors is able to produce occupation numbers that tend to 0.5 when bonds are broken, showing an interesting connection to RDMFT methods (which are known to be accurate for

strong-correlation/non-dynamic correlation effects). A proper discussion of the N-representability test is missing in this work.

8) Concerning the second ML procedure (map 2), the need to change of ML method is not clear to me. Is there any reason to prefer a regularized linear regression instead of a KRR? Also, more details about the map 2 procedure could be included in the supporting information to facilitate the reproducibility of their results. Unfortunately, the link to the Zenodo repository is not working and I could not access that data.

9) Finally, in the conclusions section the authors mention that larger molecules can be approached by combining the 1-rdms of separate fragments learned from smaller molecules. This phrase is slightly contradictory with the purpose of this work, the authors employed a bijective relation between the external (local) potential and the density to propose their method. But, combining the 1-rdms of separate fragments goes against the bijective relation. This statement should be revisited to avoid contradiction a w.r.t. the bijective relation employed that supports this work.

Considering the above-mentioned points, I cannot recommend this work for publication in its current state. For example, the ML procedures proposed in this work may have applications in MDs but only if their transferability is correct (as mentioned in point 4) it could have other applications beyond MD simulations. The spin states should already represent a challenge for this ML procedure, which (again) could limit its applicability even for MD simulations where the spin changes along the reaction path. All these issues should be checked before accepting this work for publication.

Reviewer #2 (Remarks to the Author):

This manuscript introduces a new machine-learning (ML) method capable of learning reduced density matrices. Targeting this “fundamental” quantity allows the relatively simple evaluation of various derived quantities, such as energies, forces, and dipole moments, either by direct calculation or using secondary machine-learned models (described as map 2). The accuracy of this method is demonstrated for various properties of various molecules, in general exceeding the accuracy of existing ML methods which predict fundamental properties. In particular, the ability of the method to reproduce Full-CI dynamics at a fraction of the computational cost is impressive.

Broadly speaking, the evidence in the manuscript presented supports the claim that these ML models “can generate essentially anything that a standard electronic structure code can”. Furthermore, the emphasis on the ability of this method to reproduce ab initio molecular dynamics simulations highlights the significance of the method for the broader field; as far as I am aware other ML methods which target fundamental quantities cannot yet do this.

The manuscript is missing some relevant context from the field, and suffers from occasional overstatements. In addition, the scope and applicability could be clarified, particularly by comparison to other existing methods. While the manuscript would be improved by addressing the following points, I consider them to be minor flaws.

1. The introduction includes a summary of existing methods which target fundamental quantities, such as the wavefunction and electron density. However, there exists a body of work focused on learning the Hamiltonian, which is not referenced. This should also be briefly discussed in the introduction. (See for example J. Chem. Phys. 156, 014115 (2022), Nature Computational Science volume 2, pages 367–377 (2022) and references therein).
2. The introduction correctly identifies that many ML approaches require the use of multiple ML models each targeting separate properties (e.g. one for energy surface and one for the dipole moment), and introduces this method as a means to avoid that. However, this is somewhat undermined by introducing a series of ML models for “map 2”, despite the fact that “map 2” can be directly computed. If ML is to be used for map 2, a justification should be presented why this two stage process is superior to simply learning a ML model for each property individually.
3. It is claimed that existing methods which learn the electron densities “are not sufficient[ly accurate] for their use in electronic structure methods”. This is something of an overstatement: these methods are indeed usually not accurate enough to predict forces and so cannot be used for molecular dynamics, but can predict total energies to < 1 meV/atom, which is sufficient for many chemical purposes.
4. Fig 3. shows the weakest performance of these models, but is not discussed in great detail. It appears that peak positions for all three systems are reproduced well, suggesting that the dynamics derived from the surrogate model are very accurate. However, the intensities of the peaks are not well reproduced, with a magnification of 100x required to see the lower frequency peaks in panels a) and b). This suggests that the dipole moments are not well reproduced by the surrogate model; a discussion of why this is the case would be helpful. I assume (see point 2 below) that the dipole moments used to calculate the spectra are obtained using a secondary ML model following map 2, suggesting it is this ML model which fails, rather than the map 1 model; are more accurate spectra obtained if the dipole moments are calculated directly from the predicted RDMs?
5. It would be helpful to note the limitations of this approach, particularly in reference to other existing methods. For example, it is noted that “separate training sets must be generated for each molecule and each electronic structure method targeted”; to this could be added that a training set is also limited to a specific basis set. By contrast, other methods referenced in the introduction produce models with significant extrapolative power and transferability between molecules. It also appears that this method

as written is limited to small molecules – I assume computational complexity is the reason that the benzene, 1- and 2- propanal calculations are performed using the 6-31G* basis set rather than the cc-pVTZ basis. This is hinted at in the conclusions, but readers should be aware of the limits of applicability of the method as it is currently described, and whether significant method development is required to go beyond these small molecules, especially given that other methods can already be applied to large molecules and condensed phase systems.

There are a couple of small typographical issues which could be changed to improve the clarity of the manuscript:

1. Following the definition of map 1 and map 2, γ_s is defined in opposition to γ_1 , which itself is not defined. Neither of these notations are used again. Similarly, in Tables 1 and 2 the predicted RDMs are denoted with a superscript p , whereas in the text at the end of section IIB a subscript is used.
2. There are three related methods compared in Table 2: LDAML[γ_p], LDAML[$\gamma_p + \Delta\gamma_p$] and LDAML. The first two of these are relatively clearly defined; the properties shown in table are calculated directly from the RDMs obtained using just map 1, and map 1 followed by a refinement using map 2, respectively. I assume that the final method then refers to using additional ML models for map 2, but this is never stated. This is also unclear in Fig. 1 and later in Section III: are the band gaps discussed obtained via a ML model based on the predicted 1-RDMs, or by a direct computation using the predicted 1-RDMs?
3. The word “indistinguishable” is used five times in the manuscript; sometimes this claim is supported, but other times it is not. For example, in the second paragraph of section III it is claimed that the Full-CIML energies are indistinguishable from those produced by Full-CI; this claim is not supported by Fig. 1, in which occasional small discrepancies of approximately 1-2 meV are visible by eye in panel (c). Similarly, the claim in the final paragraph of that section that “the predicted electronic structure that is essentially indistinguishable” from LDA is not supported by Fig 1d. This performance is nevertheless extremely impressive; the authors would be better served by avoiding overstatement.
4. In Fig. 2, the results for the three methods appear to be offset from one another. I assume this offset is artificially introduced by the authors for clarity, but this is not stated. If this offset is in fact an artifact of the method, it would be helpful to explain its origin.

Reviewer #3 (Remarks to the Author):

I have read with interest the article "Machine Learning Electronic Structure Methods Based On The One-Electron Reduced Density Matrix".

The article presents an approach to generating surrogate electronic structure models using machine learning (ML) together with the Hohenberg-Kohn theorems. The authors in fact use the bijective map established by the density functional theory (DFT) and reduced density matrix functional theory (RDMFT) to select input features and build physically-based models. The training data is obtained via ab initio calculations and the dataset size scales the cube of the number of vibrational degrees of freedom of the molecule. The models are then used to predict geometry optimizations, ab-initio dynamics, IR spectra, as well as expectation values of one-electron operators.

Overall, the article presents interesting results, but the ML model used in the study is too simple and system-specific. For every molecule, it is necessary to perform $(N_{\text{vib}})^3$ calculations to train the ML model, ideally with very expensive quantum chemistry methods. For a molecule as small as C₆H₆ this requires already 13824 calculations and the scaling is unfavourable.

I believe that the authors are not fully updated with the

rich literature on the topic. A very complete review can be found, e.g., in "Extending machine learning beyond interatomic potentials for predicting molecular properties" Nature Reviews Chemistry 6, p 653–672 (2022). The several examples discussed in that work prove that the application of ML has been extended to predict various chemical properties beyond energy/forces (e.g., beyond interatomic potentials). The numerical cost of the discussed advanced ML models scales linearly with the

system size and these models are extensible owing to the partitioning of the molecular structure into local chemical environments, which can be summed to reproduce a global property. Several electronic properties can be predicted thanks to these models with chemical accuracy.

Overall, while the article presents valid results, if we put it in the context of the recent literature on the topic, it does not meet the standards for publication in Nature Communications.

Subject: Reply to reviewers' comments

To Whom It May Concern:

The following major changes were implemented:

- New Figure 1 was added to the manuscript to show the energy accuracy achieved for benzene using training sets of varying sizes.
- New Figure 2 was added to the manuscript displaying the energy vs bond length for N_2 , CN^- , CO and BF demonstrating the potential transferability properties of our model.
- We now report all energies in kcal/mol, replacing our previous results which were given in meV/dof.
- Supplementary Figures S3 and S4 were added to showcase the ability of our model to approach systems featuring strong static electronic correlation (bond breaking).
- Supplementary Figure S5 displaying the time-dependent dipole moment of B3LYP^{ML} compared against B3LYP and showing that the IR oscillator strengths reported from the surrogate models would match the ones from a B3LYP simulation.
- Supplementary Table S1 displaying the RMSDs for each energy component in the total energy functional (including the HF exchange).
- Supplementary Table S2 displaying the RMSDs for the 1-rdm's occupation numbers along an AIMD trajectory.

Reply to Referee 1

- 1) In the abstract, the authors mention that the theorems of DFT and RDMFT establish a bijective map between the external potential of many-body systems and its electron density of 1-rdm. This statement is incorrect because in his paper Gilbert explicitly states that the one-to-one correspondence between the 1-rdm and a general non-local external potential is not possible. Thus, the authors should clearly mention that the bijective relation holds only for local potentials (same in the first paragraph in Pag. 2).

Reply: We agree with the reviewer and have revised the abstract accordingly. Specifically, we now only mention the theorems of DFT, which are sufficient for relating the external local potential to the one-particle reduced density matrix (1-rdm), resulting in a more concise abstract. The sentences “The theorems of density functional theory (DFT) and reduced density matrix functional theory (RDMFT) establish a bijective map between the external potential of a many-body system and its electron density or one-particle reduced density matrix. Building on this foundation, we show that machine learning can be used to generate surrogate electronic structure methods.” were changed to “The theorems of density functional theory (DFT) establish bijective maps between the local external potential of a many-body system and its electron density, wavefunction and, therefore, one-particle reduced density matrix. Building on this foundation, we show that machine learning models based on the one-electron reduced density matrix can be used to generate surrogate electronic structure methods.”

- 2) The introduction section misses a proper description of the state-of-the-art of ML procedures in the DFT and RMFT context. For example, citations to previous works are missing. The authors should cite the previous works done by M. Rupp, A. Tkatchenko, O.A. Von Lilienfeld, K. Burke, S. Lubner, M.A.L. Marques, etc. Also, the previous works using ML in the RDMFT context should be introduced and cited, in particular, Schmidt, Jonathan, Matteo Fadel, and Carlos L. Benavides-Riveros. “Machine learning universal bosonic functionals.” *Physical Review Research* 3.3 (2021): L032063 and Wetherell, Jack, et al. “Insights into one-body density matrices using deep learning.” *Faraday Discussions* 224 (2020): 265-291.

Reply: We now also cite Schmidt’s paper. The following sentence was added on page 2: “The 1-rdm is emerging as a powerful feature for describing quantum systems, even when non-local correlations are important. For example, Schmidt et al. show that approximate 1-rdm energy functionals can be constructed (even with the aid of ML) for the description of correlated bosonic systems [34].”

Wetherell et al. was already cited. However, we did not clearly describe their work in our original manuscript. The following was added on page 2 to cast our work in light of the work of Wetherell et al. “Wetherell et al. [35] considered the possibility of learning the functional \$\hat{\gamma}[n]\$, achieving accurate results for exactly solvable real-space model systems. To date, however, there have been no attempts to learn \$\hat{\gamma}[n]\$ or \$\hat{\gamma}[v]\$ (as we do here) for real systems, such as molecules.”

- 3) Minor comment: AIMD not defined.

Reply: This was corrected.

- 4) *The first ML procedure (labeled as map 1) aims to avoid the computation of the 1-rdm from an expensive electronic structure calculation. The transferability of this procedure raises my major concern related to this work. I understand that for each system in table II the authors performed a KRR using some samples (N_{samples}). This procedure will definitely lower the computational cost during a MD simulation. But, if one uses the cc-pVTZ basis set, for example, and performs the KRR regression for N_2 with LDA (as the authors did for H_2O , CO_2 , etc.). Would the authors be able to predict the 1-rdm for a given geometry of CO by building the corresponding potential in the GTO basis and evaluating it in the KRR obtained for N_2 ? (Note: N_2 and CO in the cc-pVTZ basis set produce the same number of GTOs; also, both systems contain 14 electrons). The transferability test of the map 1 ML procedure should be addressed and discussed (also including other examples).*

Reply: In the original manuscript, we intentionally did not focus on testing the transferability of our ML models. However, in response to the reviewer’s suggestion, we conducted three additional tests. In Case 1, we trained a model on N_2 data and predicted CO; however, the resulting CO electronic structure exhibited a symmetric pattern, which is unphysical. In Case 2, we trained the model on CO data and predicted N_2 ; however, the predicted N_2 electronic structure showed a small dipole. In Case 3, we trained the model on a combined set of N_2 and CO data and then predicted N_2 , CO, BF and the anion CN^- ; in this case, the results indicate that BF and CN^- can be predicted with a high degree of accuracy, even though their structures were not included in the training set. This is summarized in the new Figure 2 of the main text which for convenience is reproduced in the following page. While these results suggest that our model may be transferable to large classes of molecular systems, we emphasize that they are not statistically significant. Therefore, a comprehensive assessment of the transferability of our proposed ML models will need to be conducted in future works. We now state on page 5: “While a thorough analysis of the transferability properties of our ML models is beyond the scope of this work, we tested the ability of the model trained on two molecular species, CO and N_2 , together to predict two systems not included in the training set, the CN^- anion and BF. Training and test molecules are isoelectronic and share the same number of AOs within the cc-pVTZ basis set which was chosen for this test. The results are displayed in Figure 2, and show that the ML model can predict the energy of CN^- and BF vs bond length quite well with an R^2 value of 0.981 and 0.975, respectively. For CO and N_2 (which are part of the training set) the R^2 score is 0.9999.”

Figure 2: Total energy as a function of bond length for the isoelectronic series N_2 , CO , CN^- and BF . The training set is composed only of N_2 and CO geometries with bond lengths indicated by the shadow region. B3LYP results are shown by scatter points, and our surrogate model results are represent by lines.

- 5) My second concern is related to spin-states. For different spin states the external potential represented in the GTO basis set is the same, but the 1-rdm are (in general) very different. This limitation of their ML procedure should be clearly mentioned in the manuscript.

Reply: We agree with the reviewer that a mention of degenerate states is needed. We now clearly state in the manuscript on page 2: “Caution must be used, as the bijective nature of the maps considered in the case of RDMFT can only be guaranteed for non-degenerate ground states. When spin systems are considered, additional constraints and features for the ML model for the 1-rdm would be required [27, 28].”

- 6) The authors reported in Table II the RMSDs for some quantities. The larger errors in the Kinetic energy w.r.t. total Energies make me consider that the non-locality of the 1-rdm might be hard to reproduce by their ML procedure. For DFT, the authors could include, at least in the supporting information, all the energy components (i.e. the Kinetic energy, the interaction with the external potential, V_{Hartree} , V_{xc} , etc.). Actually, for the Hartree-Fock tests, the exchange energy could also serve as an excellent tool to evaluate the quality of the non-local character of the predicted 1-rdms.

Reply: We have added a new table (Table S1) to the supplementary materials, which presents all the terms in the energy functional of LDA for all the molecules used in this study. Additionally, we have computed the HF exchange energy from the LDA and LDA^{ML} 1-rdms

and included them in Table S1 to address the question about HF exchange. The RMSDs for the various terms in the energy functional are similar in size to the ones presented before for the non-interacting kinetic energy. However, we acknowledge that there is a degree of error cancellation, as the RMSD for the individual components of the energy functional is larger than the RMSD for the total energy. The results regarding the HF exchange indicate that our ML model correctly predicts the non-locality of the 1-rdm.

We inserted the following sentence on page 5: “In Table S1, we present the RMSDs for individual terms in the energy functional. It is observed that the RMSDs of these separate terms are larger compared to the RMSD for the total energy. This suggests that the model is benefiting from some degree of error cancellation. Notably, however, our model can accurately predict HF exchange energies using the predicted 1-rdms (with an RMSD of less than 0.2 kcal/mol).”

- 7) *In line with the previous point, it is important to test the N-representability of reduced density matrices. In the KS-DFT/HF cases, the 1-rdm must correspond to that of a single-determinant wave function. Thus, upon diagonalization of the 1-rdm (probably diagonalizing $S^{1/2}PS^{1/2}$) the authors could produce the occupation numbers and discuss whether or not their ML procedure is able to produce predicted 1-rdms that correspond to a single-determinant wave function. At the FCI level, similar tests should be performed considering that occupation numbers should yield between 0 and 1 (see C. Garrod and J. K. Percus *J. Math. Phys.* 5, 1756 (1964) <https://aip.scitation.org/doi/pdf/10.1063/1.1704098>). This test could reveal if the ML procedure proposed by the authors is able to produce occupation numbers that tend to 0.5 when bonds are broken, showing an interesting connection to RDMFT methods (which are known to be accurate for strong-correlation/non-dynamic correlation effects). A proper discussion of the N-representability test is missing in this work.*

Reply: We are grateful to the reviewer for bringing up the issue of N-representability which we only partially described in the original manuscript. We agree that the issue requires additional discussion and analysis. To briefly summarize, the predicted 1-rdms may be approximately N-representable if left unpurified. The new Table S2 in the supplementary materials shows that the RMSDs for the occupation numbers of unpurified 1-rdms range from approximately 10^{-5} to 10^{-3} . While unpurified 1-rdms can be used for stable AIMD trajectories and geometry optimizations for small molecules like water, longer simulations for larger molecules are limited to short times (less than 1 ps). To overcome this limitation, we implemented a purification step ($\text{diag}(S^{1/2}PS^{1/2}) = 2$ or $= 0$). We acknowledge that this purification step involves diagonalization, which scales cubically with system size. However, we note that there are linear-scaling, iterative methods available to achieve 1-rdm purification without diagonalization (see, e.g., <https://onlinelibrary.wiley.com/doi/pdf/10.1002/qua.25048>, <https://aip.scitation.org/doi/pdf/10.1063/1.4943213>).

We now add Table S2 in the supplementary materials and a paragraph on page 5 discussing N-representability: “After testing (see Table S2 for RMSDs of the 1-rdm’s occupation numbers), we observed that strict non-interacting N-representability of the 1-rdms is crucial for accessing stable, long AIMDs for molecules that are larger than triatomics. To achieve this, we compute the occupation numbers of the predicted 1-rdms and enforce aufbau occupation. However, we omit this purification step when learning full configuration interaction (Full-CI) 1-rdms. In section III, we will also test the ability of the model 1-rdms to predict fractional occupations

upon bond breaking.”

We investigated the ability of our ML model to predict fractional occupations by examining the dissociation of a water molecule. Specifically, we stretched either one bond ($\text{H}_2\text{O} \rightarrow \text{OH} + \text{H}$) or two bonds ($\text{H}_2\text{O} \rightarrow \text{O} + \text{H}_2$), and added two geometries to the training set associated with the systems $\text{OH} + \text{H}$ and $\text{O} + \text{H}_2$ to provide information about the dissociation limit to the ML model. The results of Full-CI^{ML} and Full-CI along a bond dissociation path are shown in Figures S3 and S4 (for convenience, we reproduce them below), which display the predicted occupation numbers and dipole moment components. We found that in general the model 1-rdms behave physically, producing fractional occupations of the frontier orbitals. However, for the case of double bond-breaking (with dissociation limit $\text{O} + \text{H}_2$), the predicted occupations become fractional a bit too early. Additionally, while the dipole moment components are qualitatively reproduced, noticeable deviations are present.

To address this comment, we now include the following paragraph on page 6: “To evaluate the performance of our model, Full-CI^{ML} 1-rdms in predicting fractional occupations upon bond breaking, we extended the water molecule training set with two additional dissociation geometries, $\text{H}_2\text{O} \rightarrow \text{OH} + \text{H}$ and $\text{H}_2\text{O} \rightarrow \text{O} + \text{H}_2$. As shown in Figures S3 and S4, the model 1-rdms are able to smoothly interpolate between aufbau occupation states at the equilibrium geometry and fractional occupations at the dissociation limits. However, for the Full-CI occupations of $\text{O} + \text{H}_2$, a sharp transition is observed associated with the breaking of the O-H bonds and formation of the H-H bond, which is only smoothly interpolated by the model 1-rdms. Despite this limitation, the dipole moment vectors in the two dissociation limits are qualitatively reproduced by the model. Overall, these results suggest that bond breaking can be qualitatively described by providing the ML model with critical information about the dissociation limits. To achieve a more accurate description of bond breaking, we anticipate that increasing the size of the training set will lead to progressively more accurate model 1-rdms.”

Figure S3: The occupation numbers and dipole moment components from Full-CI^{ML} predicted γ^p and Full-CI γ for single bond stretching ($\text{H}_2\text{O} \rightarrow \text{OH} + \text{H}$) of H_2O .

Figure S4: The occupation numbers and dipole moment components from Full-CI^{ML} predicted γ^p and Full-CI γ for double bond stretching ($\text{H}_2\text{O} \rightarrow \text{O} + \text{H}_2$) of H_2O .

- 8) Concerning the second ML procedure (map 2), the need to change of ML method is not clear to me. Is there any reason to prefer a regularized linear regression instead of a KRR? Also, more details about the map 2 procedure could be included in the supporting information to facilitate the reproducibility of their results. Unfortunately, the link to the Zenodo repository is not working and I could not access that data.

Reply: We tested several regression methods, including KRR and regularized linear regressions (rLR), and determined that rLR performed the best for map 2 for all systems considered. The thorough analysis of other regressions (including ones based on neural networks) is subject of current investigations. We rephrased one sentence in the manuscript which now reads: “As we have seen, KRR is employed for γ -learning. For $\gamma + \delta$ -learning, after some testing, we chose to use a regularized linear regression.”

We apologize for the inoperable link for the supplementary documents in our original submission which we have now corrected. The supplementary materials are available on Zenodo at the link: <https://zenodo.org/record/7661223>.

- 9) Finally, in the conclusions section the authors mention that larger molecules can be approached by combining the 1-rdms of separate fragments learned from smaller molecules. This phrase is slightly contradictory with the purpose of this work, the authors employed a bijective relation between the external (local) potential and the density to propose their method. But, combining the 1-rdms of separate fragments goes against the bijective relation. This statement should be revisited to avoid contradiction a w.r.t. the bijective relation employed that supports this work.

Reply: The point raised by the reviewer is interesting. What we meant is that one can learn 1-rdms of molecular fragments and use them as good initial guesses for the 1-rdms of larger molecules. These initial guesses would not include inter-fragment off-diagonal blocks, but those can also be estimated based on overlaps of fragment orbitals. Nonetheless, the deviation between the initial guess 1-rdm and the true 1-rdm of the full system would still need to be machine learned following the approach described in this work. We rephrased the sentence in question which now reads: “To approach large molecules, initial guesses of

the 1-rdms can be constructed combining the 1-rdms of separate molecular fragments learned from smaller molecules. Their coupling can then be learned separately.”

Reply to Referee 2

- 1. *The introduction includes a summary of existing methods which target fundamental quantities, such as the wavefunction and electron density. However, there exists a body of work focused on learning the Hamiltonian, which is not referenced. This should also be briefly discussed in the introduction. (See for example J. Chem. Phys. 156, 014115 (2022), Nature Computational Science volume 2, pages 367–377 (2022) and references therein).*

Reply: We added the following paragraph on page 1-2: “It is worth noting that there are various developing and complementary techniques that involve learning Hamiltonian matrices with respect to localized orbitals using symmetrized 2- [13] or N-center [14] representations, or deep neural network representation [15]. These methods share some of the underlying motivations of our work. For further reading on this and related topics, we recommend a recent comprehensive review [16].”

- 2. *The introduction correctly identifies that many ML approaches require the use of multiple ML models each targeting separate properties (e.g. one for energy surface and one for the dipole moment), and introduces this method as a means to avoid that. However, this is somewhat undermined by introducing a series of ML models for “map 2”, despite the fact that “map 2” can be directly computed. If ML is to be used for map 2, a justification should be presented why this two stage process is superior to simply learning a ML model for each property individually.*

Reply: We appreciate the reviewer’s observation that map 2 can be used to target specific observables and note that it is similar in spirit to typical workflows. However, there are several distinctions that should be made. Firstly, while map 2 may not be strictly necessary for mainstream DFT methods or for any observable depending only on the 1-rdm, it is required for Full-CI since the energy functional is not purely a functional of the 1-rdm. Theorems of RDMFT tell us that the functional $E[\gamma]$ exists, but its expression in terms of the 1-rdm is unknown. Furthermore, ongoing research in our lab is developing an ML model similar to the one presented in this manuscript but for the 2-rdm. When employing the 2-rdm, there is no need to perform targeted learning of energy and forces, even in the case of wavefunction-based methods. To address this comment, we have modified a sentence on page 3, as follows: “When considering mean field methods, map 2 can either be learned or directly computed from predicted 1-rdms. In the following sections, we will consider these options and show that they both lead to equally successful outcomes (vide infra). For post-HF wavefunction-based methods, map 2 must be learned as there are no pure functionals of the 1-rdm that deliver energy and forces for these methods.”

- 3. *It is claimed that existing methods which learn the electron densities “are not sufficient[ly accurate] for their use in electronic structure methods”. This is something of an overstatement: these methods are indeed usually not accurate enough to predict forces and so cannot be used for molecular dynamics, but can predict total energies to <1 meV/atom, which is sufficient for many chemical purposes.*

Reply: The sentence in question was modified to read: “While energies obtained using these methods can be accurate to around 1 kcal/mol, the model densities usually have deviations from the target on the order of a few percent. However, even with these small deviations, the model densities cannot be used directly to evaluate ionic forces in a DFT algorithm. This is because self-consistent electron densities must be converged to within much tighter thresholds for accurate force calculations.”

We also refer the reviewer to our answer to comment 1 of Reviewer 3 for additional information on aspects related to this comment.

- 4. *Fig 3. shows the weakest performance of these models, but is not discussed in great detail. It appears that peak positions for all three systems are reproduced well, suggesting that the dynamics derived from the surrogate model are very accurate. However, the intensities of the peaks are not well reproduced, with a magnification of 100x required to see the lower frequency peaks in panels a) and b). This suggests that the dipole moments are not well reproduced by the surrogate model; a discussion of why this is the case would be helpful. I assume (see point 2 below) that the dipole moments used to calculate the spectra are obtained using a secondary ML model following map 2, suggesting it is this ML model which fails, rather than the map 1 model; are more accurate spectra obtained if the dipole moments are calculated directly from the predicted RDMS?*

Reply: The dipole moments employed in the IR spectra dynamics are very close to those generated by the corresponding QM approach. All the dipole moments are calculated directly from the predicted 1-rdms. The deviations in peak intensity, as noted by the reviewer, can be explained by two factors. Firstly, since B3LYP^{ML} cannot provide results superior to the parent B3LYP, and even though B3LYP is highly accurate, it is not infallible and has been observed to produce deviated spectra from experiments, both from harmonic calculations and MD simulations [10.1021/acs.jctc.0c00126,10.1002/adts.202100293]. Secondly, we have not conducted a comprehensive analysis of the IR spectra convergence with respect to the number of MD trajectories employed.

To address this comment, we added the new Figure S5 in the supplementary materials and rephrased the paragraph on page 7: “The model successfully reproduces the peak positions of the experimental spectra and follows the correct trend for peak intensities. However, due to the flexible nature of the molecules, particularly 1-propanol, a greater number of snapshots and longer trajectories may be necessary to obtain a more accurate comparison to the experiment. Additionally, the use of the B3LYP xc functional, which is an approximation, introduces errors in the predicted IR spectra, affecting both the frequency positions and the oscillator strengths [44, 45]. Our B3LYP^{ML} model excellently reproduces the B3LYP dipoles along one of the AIMD trajectories used for the computation of the IR spectra (see, e.g., Figure S5), showing that the deviations from the experiment are likely due to the approximations involved in the B3LYP xc functional.”

- 5) *It would be helpful to note the limitations of this approach, particularly in reference to other existing methods. For example, it is noted that “separate training sets must be generated for each molecule and each electronic structure method targeted”; to this could be added that a training set is also limited to a specific basis set. By contrast, other methods referenced in the*

introduction produce models with significant extrapolative power and transferability between molecules. It also appears that this method as written is limited to small molecules – I assume computational complexity is the reason that the benzene, 1- and 2- propanal calculations are performed using the 6-31G basis set rather than the cc-pVTZ basis. This is hinted at in the conclusions, but readers should be aware of the limits of applicability of the method as it is currently described, and whether significant method development is required to go beyond these small molecules, especially given that other methods can already be applied to large molecules and condensed phase systems.*

Reply: Comment 4 of Reviewer 1 touched on a similar topic. To address this comment, we inserted text after the quoted sentence by the reviewer on page 4: “Separate training sets must be generated for each molecule and electronic structure method chosen, including any method-specific settings such as basis sets, charge, and spin multiplicity. This aspect of our method can be seen as both an advantage and a limitation. On one hand, it allows for flexibility in choosing the electronic structure method best suited for a particular application. On the other hand, it means that in principle, a new ML model should be trained for each combination of settings. However, our results in the next section show that the limitations of the method are mitigated by its potentially good transferability.”

The mentioned results are given by the new calculations performed and presented in our reply to comment 4 of Reviewer 1.

There are a couple of small typographical issues which could be changed to improve the clarity of the manuscript:

Reply: We have corrected the issues as follows:

- 1. Following the definition of map 1 and map 2, γ_s is defined in opposition to γ_1 , which itself is not defined. Neither of these notations are used again. Similarly, in Tables 1 and 2 the predicted RDMs are denoted with a superscript p , whereas in the text at the end of section IIB a subscript is used.

Reply: We have removed the definitions that were used only once.

- 2. There are three related methods compared in Table 2: $LDA^{ML}[\gamma^p]$, $LDA^{ML}[\gamma^p + \Delta\gamma^p]$ and LDA^{ML} . The first two of these are relatively clearly defined; the properties shown in table are calculated directly from the RDMs obtained using just map 1, and map 1 followed by refinement using map 2, respectively. I assume that the final method then refers to using additional ML models for map 2, but this is never stated. This is also unclear in Fig. 1 and later in Section III: are the band gaps discussed obtained via a ML model based on the predicted 1-RDMs, or by a direct computation using the predicted 1-RDMs?

Reply: Referee is correct about the definition of the notations. $LDA^{ML}[\gamma^p]$ and $LDA^{ML}[\gamma^p + \Delta\gamma^p]$ indicate that all properties are calculated directly from the -rdm from map 1 and map

2, respectively. LDA^{ML} is equivalent to $\text{LDA}^{\text{ML}}[\gamma^p + \Delta\gamma^p]$, except energy and forces are predicted directly using map 2. Therefore, the band gaps and orbitals in Figure 2 (which was Figure 1 in the original manuscript) and Section III are computed by 1-rdms from map 2. We have added following on page 4: “To summarize, we introduced three models: $\text{QM}^{\text{ML}}[\gamma^p]$, $\text{QM}^{\text{ML}}[\gamma^p + \Delta\gamma^p]$, and QM^{ML} , where QM can be any quantum mechanical method that yields the 1-rdm. $\text{QM}^{\text{ML}}[\gamma^p]$ and $\text{QM}^{\text{ML}}[\gamma^p + \Delta\gamma^p]$ denote methods where all properties are calculated directly from the 1-rdm from map 1 and map 2, respectively. QM^{ML} is equivalent to $\text{QM}^{\text{ML}}[\gamma^p + \Delta\gamma^p]$, except energy and forces are predicted directly using map 2.”

We added the following in the captions of Figures 2 and 5: “For LDA^{ML} , the HOMO-LUMO gaps and orbitals are computed with predicted 1-rdms from map 2.”

- 3. The word “indistinguishable” is used five times in the manuscript; sometimes this claim is supported, but other times it is not. For example, in the second paragraph of section III it is claimed that the Full- CI^{ML} energies are indistinguishable from those produced by Full-CI; this claim is not supported by Fig. 1, in which occasional small discrepancies of approximately 1-2 meV are visible by eye in panel (c). Similarly, the claim in the final paragraph of that section that “the predicted electronic structure that is essentially indistinguishable” from LDA is not supported by Fig 1d. This performance is nevertheless extremely impressive; the authors would be better served by avoiding overstatement.

Reply: We agree with the reviewer and have amended our descriptions of the predicted quantities replacing ‘indistinguishable’ with ‘very close’ or ‘extremely close’.

- 4. In Fig. 2, the results for the three methods appear to be offset from one another. I assume this offset is artificially introduced by the authors for clarity, but this is not stated. If this offset is in fact an artifact of the method, it would be helpful to explain its origin.

Reply: The referee is correct. We have added a sentence in the caption of Figure 4 (originally Figure 2): “To aid visualization, energies for LDA^{ML} , $\text{LDA}^{\text{ML}}[\gamma^p + \Delta\gamma^p]$ are shifted by 4 and 8 kcal/mol, respectively for panel (a) and by 2 and 4 kcal/mol, respectively for panel (b).”

Reply to Referee 3

- 1) Overall, the article presents interesting results, but the ML model used in the study is too simple and system-specific. For every molecule, it is necessary to perform $(N_{\text{vib}})^3$ calculations to train the ML model, ideally with very expensive quantum chemistry methods. For a molecule as small as C6H6 this requires already 13824 calculations and the scaling is unfavourable.
- 2) I believe that the authors are not fully updated with the rich literature on the topic. A very complete review can be found, e.g., in "Extending machine learning beyond interatomic potentials for predicting molecular properties" *Nature Reviews Chemistry* 6, p 653–672 (2022). The several examples discussed in that work prove that the application of ML has been extended to predict various chemical properties beyond energy/forces (e.g., beyond interatomic potentials). The numerical cost of the discussed advanced ML models scales linearly with the system size and these models are extensible owing to the partitioning of the molecular structure into local chemical environments, which can be summed to reproduce a global property. Several electronic properties can be predicted thanks to these models with chemical accuracy.

Reply: We address both comments as they are closely related. Generally, ML is a data-intensive method, and the size of the data set required for training strongly depends on the target systems and processes. The question at hand is the size of the configuration space needed to represent the electronic structure of a molecular system. While we cannot comment on the perceived simplicity of our approach by this reviewer, we can provide an example related to the accuracy of our model with smaller training set sizes than reported in the manuscript. New Figure 1, now included in the main text, demonstrates that if chemical accuracy is sought in the prediction of the energy, it is sufficient to employ 30 structures for benzene. A one-order-of-magnitude improvement over chemical accuracy requires 100 structures, while achieving nearly three orders of magnitude better accuracy than "chemical accuracy" necessitates the 13824 structures reported in the manuscript. Our objective is not chemical accuracy, but much more accurate energies than that. The reason for this requirement is that the target of our model is not merely the energy, but rather the electronic structure exemplified by the 1-rdm of the system. From the 1-rdm we can calculate or predict energies, forces, and anything else that a conventional quantum chemistry code can deliver. We now also include a reference to the mentioned paper (which has been cited also in response to comment 1 of Reviewer 2), included Figure 1 and the following paragraph on page 4:

"The goal of our ML model is to achieve accuracies beyond chemical accuracy, as our model aims to predict the 1-rdm with an accuracy that matches the self-consistent field (SCF) densities produced by conventional quantum chemistry codes. SCF densities are typically converged to energy thresholds of 10^{-5} or 10^{-6} Hartree, which is two-to-three orders of magnitude more accurate than chemical accuracy. In Figure 1, we show the energy accuracy achieved for the total energy of benzene using training sets with varying numbers of structures. Chemical accuracy is already achieved with a training set size of 30 structures, and a one-order-of-magnitude improvement can be achieved by increasing the training set size to 100. However, to achieve accuracies three orders of magnitude below chemical accuracy and match the accuracy of conventional SCF-derived densities, larger training sets are required, as reported above."

FIG. 1: The accuracy of the $\text{LDA}^{\text{ML}}[\gamma^p]$ and $\text{LDA}^{\text{ML}}[\gamma^p + \Delta\gamma^p]$ model as a function of training set size for the benzene molecule. See caption to Table II for details regarding the RMSD calculation.

REVIEWERS' COMMENTS

Reviewer #1 (Remarks to the Author):

Referee report

The authors have adequately addressed my comments and suggestions. And, they have also included statements in the article that show the *benefits and drawbacks* of the proposed methods. More specifically, my major concern about the transferability of this method among systems has been answered. As the authors have shown (and also added to the rebuttal letter), the transferability even among isoelectronic systems like N₂ and CO is a hard task for their approach. Nevertheless, they have clearly clarified this issue in the current version of the article. Similarly, I consider that the analysis performed for the occupation numbers and N-representability has improved the quality of this work. Thus, I am keen to recommend it for publication.

I have only seen a couple of minor errors in the document in its current state:

a) In Pag. 3 in the phrase "We note that when DFT methods are considered, the KS 1-rdm the KS 1-rdm is targeted." there is repeated "the KS 1-rdm".

b) In Pag. 2, "achieving accurate results for exactly solvable real-space model systems. achieving good results for model systems." The phrase is not completely clear to me.

Reviewer #2 (Remarks to the Author):

I am satisfied that the modifications to the manuscript and Supporting Information made by the authors have effectively addressed both my concerns and those raised by the other referees, and recommend publication.

Reviewer #3 (Remarks to the Author):

The revised manuscript includes more results and clarifies several points. I am still not convinced about the practical utility of this method, but I find the idea and the discussion valuable and I recommend to publish. In the introduction, the discussion of the state of the art of machine learning in DFT and RDMFT is still rather limited.

Point-by-point response to referees

Reviewer #1

1. In Pag. 3 in the phrase "We note that when DFT methods are considered, the KS 1-rdm the KS 1-rdm is targeted." there is repeated "the KS 1-rdm".

REPLY: We have corrected the phrase.

2. In Pag. 2, "achieving accurate results for exactly solvable real-space model systems. achieving good results for model systems." The phrase is not completely clear to me.

REPLY: We have edited the phrase for clarity. It now reads: "Wetherell et al.³⁵ considered the possibility of learning the functional $\hat{\gamma}[n]$ from exactly solvable real-space model systems achieving promising results."